# Integrated $CO_2$ capture-fixation chemistry via interfacial ionic liquid catalyst in laminar gas/liquid flow

Niraj K. Vishwakarma[1,*], Ajay K. Singh[1,2,*], Yoon-Ho Hwang[1], Dong-Hyeon Ko[1], Jin-Oh Kim[1], A. Giridhar Babu[1] & Dong-Pyo Kim[1]

Simultaneous capture of carbon dioxide ($CO_2$) and its utilization with subsequent work-up would significantly enhance the competitiveness of $CO_2$-based sustainable chemistry over petroleum-based chemistry. Here we report an interfacial catalytic reaction platform for an integrated autonomous process of simultaneously capturing/fixing $CO_2$ in gas–liquid laminar flow with subsequently providing a work-up step. The continuous-flow microreactor has built-in silicon nanowires (SiNWs) with immobilized ionic liquid catalysts on tips of cone-shaped nanowire bundles. Because of the superamphiphobic SiNWs, a stable gas–liquid interface maintains between liquid flow of organoamines in upper part and gas flow of $CO_2$ in bottom part of channel. The intimate and direct contact of the binary reagents leads to enhanced mass transfer and facilitating reactions. The autonomous integrated platform produces and isolates 2-oxazolidinones and quinazolines-2,4(1H,3H)-diones with 81–97% yields under mild conditions. The platform would enable direct $CO_2$ utilization to produce high-valued specialty chemicals from flue gases without pre-separation and work-up steps.

[1] National Center of Applied Microfluidic Chemistry, Department of Chemical Engineering, POSTECH (Pohang University of Science and Technology), Pohang 37673, Korea. [2] Division of Natural Product Chemistry, CSIR-Indian Institute of Chemical Technology, Hyderabad 500007 India. * These authors contributed equally to this work. Correspondence and requests for materials should be addressed to D.-P.K. (email: dpkim@postech.ac.kr).

One of the biggest factors of global warming is tens of gigatons emission per year of carbon dioxide ($CO_2$) to the environment[1,2]. Therefore, extensive efforts have been made to develop practical $CO_2$ capture and sequestration technologies including absorption, adsorption, cryogenics, and membrane using aqueous amines and functional materials such as metal-organic framework, zeolite, metal oxides and so on[3–5]. However, these technologies have some inherent drawbacks including low $CO_2$ capacity, solvent loss, equipment corrosion, high cost for regeneration[6]. On the other hand, the $CO_2$-based sustainable chemistry that converts the captured $CO_2$ to commodity chemicals such as formic acid[7,8] and methanol[9,10] has suffered from inherent thermodynamic stability and low reactivity of $CO_2$ even under high $CO_2$ pressure[11] and harsh reaction conditions[12–14], failing to overcome the competitiveness of the petroleum-based chemistry.

Ideally, it is desirable to simultaneously accomplish $CO_2$ separation and reaction in one vessel. One way of accomplishing the task is to have $CO_2$ in a gas mixture reacted with a reactant to produce a commodity chemical. For the purpose, an efficient catalyst is needed. It has come to our attention that a catalysis system based on organometallic[10,15–17] and 1,8-diaza-bicyclo[5.4.0]undec-7-ene (DBU)[18] is efficient in utilizing $CO_2$ for the synthesis of heterocyclic drug compounds. In particular, these ionic liquids (ILs) containing amino-function group have excellent $CO_2$ absorption capacity in addition to low vapour pressure, good thermal stability, high polarity and non-toxicity, and act as good solvents in organic synthesis. However, the highly viscous ionic liquid suffered from low gas-to-liquid mass transfer in addition to insufficient surface-to-volume ratio and heat transfer of conventional batch reactor. Therefore, the limited process efficiency required large amounts of DBU-based catalyst (2–6 equiv.) and long reaction times (12–24 h) to attain reasonable chemical performance[19,20]. Furthermore, the purification of expensive DBU-based IL catalyst for recovery and reuse is highly laborious.

In this report, we present an interfacial catalytic reaction platform for simultaneously capturing and fixing $CO_2$ in gas–liquid laminar flow. Furthermore, a subsequent work-up step is provided to isolate the desired synthesized products such as 2-oxazolidinones and quinazolines-2,4($1H$,$3H$)-diones. The continuous-flow microreactor system has a built-in upper panel and a bottom panel that is a silicon substrate on which superamphiphobic Silicon nanowires (SiNWs) were fabricated. The liquid of organoamines flows through the upper part of the channel and the gas containing $CO_2$ flows through the bottom part, which operates without cross-penetration between binary phases because of the super-amphiphobicity. Therefore, a distinct gas–liquid interface is maintained throughout the channel, although the liquid phase is above the gas phase. The DBU-based IL catalysts are immobilized on the tips of the bundles of SiNWs in the form of a thimble.

Enhanced mass transfer and the specific positioning of catalysts at the binary interface facilitate the reactions between gas and liquid regents in laminar flow, enabling 81–97% yields under mild conditions. The immobilized catalyst allowed repeated use of the catalyst. Furthermore, facile isolation of product from the mixture was achieved by autonomous work-up step via solvent incorporation techniques in the integrated microfluidic system. This hybrid and integrated process via *in situ* utilization of $CO_2$ from pure or diluted $CO_2$ resources is demonstrated for direct production of high-value specialty chemicals and purification of natural gas for subsequent use, which would be useful for a practical on-site $CO_2$-based sustainable chemistry.

## Results

**Superamphiphobic SiNWs and immobilized catalyst.** The superamphiphobic SiNWs on the bottom panel of the reactor was fabricated by Ag-assisted anisotropic etching of silicon wafer (100) according to the reported methods[21,22], and the DBU-ILs were selectively positioned on the tips of cone-shaped bundles of SiNWs in the form of a thimble (Supplementary Fig. 1). The length and diameter of SiNWs were controlled by etching time and Ag-nanoparticle loading time, respectively, to obtain a length of 70–75 μm with a diameter 150–300 nm. The scanning electron microscopy (SEM) images of the top and cross-sectional views of SiNWs are shown in Supplementary Fig. 2. The cone-shaped SiNWs were decorated with $SiO_2$ nanoparticles followed by fluorination to yield a hierarchical structure and rough surface, which is required to achieve a superamphiphobic interface[23]. To immobilize DBU-ILs selectively only on the bundle tips of superamphiphobic SiNWs, the tip parts (2–3 μm) of SiNWs were coated temporarily by a high viscosity molten wax at 80 °C while the uncoated bottom parts were maintained bare for subsequent fluorination. The selectively fluorinated SiNWs by trichloroper-fluorooctylsilane showed high static contact angles (CAs) of 128° and 153° for DMSO and water, respectively, as seen in Supplementary Fig. 3a,b, which are somewhat lower than the CAs of 155° and 164° for the entirely fluorinated SiNWs without wax protection of the thimbles[22]. The CA required for a stable laminar flow with high liquid-repelling capability was achieved by optimizing four parameters: (1) silver-nanoparticles loading time of 4.5 min, (2) etching time for 6 h, (3) $SiO_2$ decoration concentration at 24 mM of TEOS and (4) fluorination time for 6 h at 55 °C, as plotted in Supplementary Fig. 3c.

For immobilization of IL catalyst, the thimble parts of SiNWs were cleanly de-protected from the wax to expose the Si-OH surface that was chemically modified to -SH group by sol–gel reaction with (3-mercaptopropyl)trimethoxysilane. Then, the microreactor with built-in SiNWs was fabricated by tightly bonding the bottom silicon panel with SiNWs clusters in the serpentine channel (83 cm length × 500 μm width × 70 μm height) with the surface-modified PDMS panel (80 cm length × 500 μm width × 30 μm height), which has with two inlets and two outlets for separate control of $CO_2$ gas and liquid samples, as seen in Fig. 1 and Supplementary Fig. 4a,b. The inner surface of the PDMS channel was modified by photo/thermo curable pre-cera-mic allylhydridopolycarbosilane (AHPCS) protecting layer (~5 μm thickness) as a precursor of silicate glass via hydrolysis to avoid the swelling due to organic solvents as reported (Supplementary Figs 4a and 5)[24,25].

To immobilize DBU-ILs on the tips of cylindrical bundles of SiNWs via ultraviolet-initiated thiol-ene click chemistry[26], the -SH-modified SiNWs-PDMS microreactor was filled with 1 wt% 6-allyl-1,8-diazabicyclo[5.4.0]undec-7-ene (A-DBU, detailed pro-cedure of synthesis and characterization in Supplementary Methods, section 2.2 and Supplementary Figs 6–10) solution in THF and irradiated with ultraviolet for 30 min at room temperature. It was found that ~0.86 mg cm$^{-2}$ of A-DBU was immobilized on the thimbles of SiNWs. The total amount of immobilized catalyst was quantified by comparing the initial and final concentrations of A-DBU after repeating the infusion three times (details in Supplementary Methods). The DBU was also immobilized on the -OH surface of the PDMS panel modified with AHPCS employing the same procedure, which was confirmed by FTIR spectrum (Supplementary Fig. 11).

The immobilized DBU was turned to its corresponding ILs by simply flowing a weak base of either 2-methylimidazole (MIm) or 2,2,2-trifluoroethanol (TFE) solution in THF to form [HDBU$^+$][MIm$^-$] and [HDBU$^+$][TFE$^-$] catalyst as the thimbles of SiNWs (details in Supplementary Figs 12 and 13

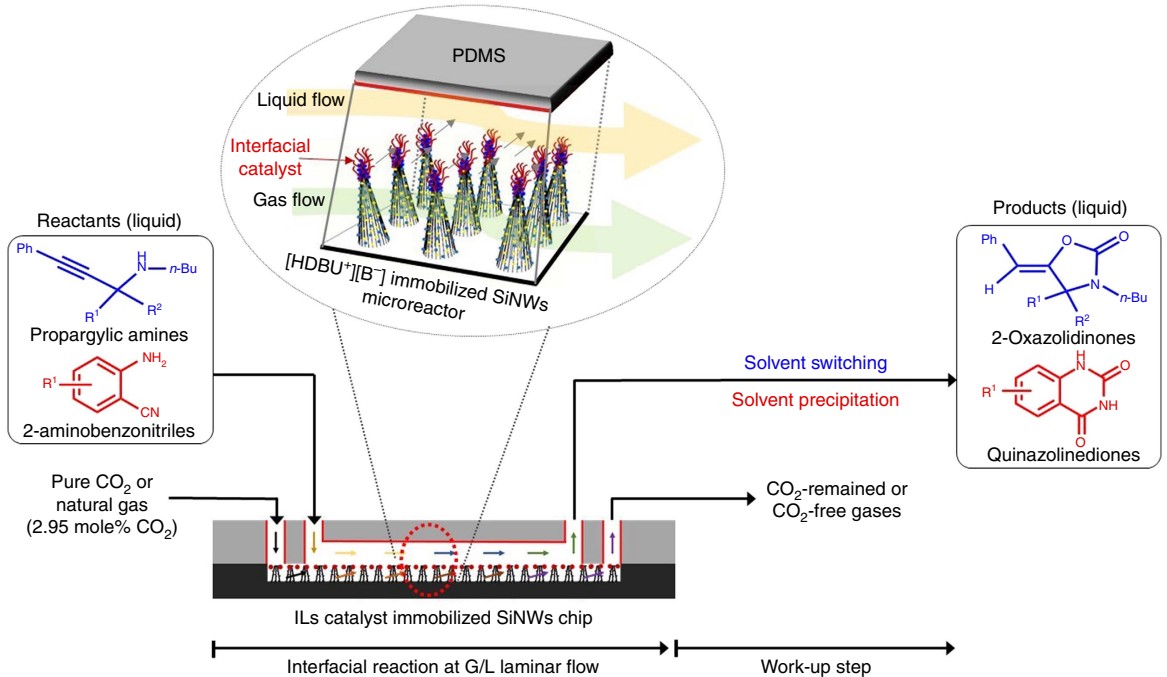

**Figure 1 | Interfacial catalytic reaction platform.** An integrated gas–liquid microfluidic system and overall process for capture and fixation of $CO_2$ from pure or diluted $CO_2$ resources to synthesize two heterocyclics of 2-oxazolidinones and quinazolines-2,4(1H,3H)-diones, and consecutive work-up for facile product isolation.

and Supplementary Methods), respectively. The cation and anion pair has a synergic effect for the catalysis[19,20]. It is noted that an appropriate solvent must be chosen to assure a CA that is high enough for maintaining a stable laminar flow over the DBU-ILs immobilized SiNWs. Based on the solubility of reactants and products, THF, 1,4-dioxane and dimethylsulfoxide (DMSO) were initially screened from various solvents (Supplementary Table 1). Eventually, the DMSO with high surface tension was found to be a suitable solvent, rather than THF and 1,4-dioxide that have a low surface tension.

The DBU-ILs immobilization on the SiNWs did not cause any change in morphology from the original SiNWs according to the cross-sectional and top views of high-resolution SEM as seen in Supplementary Fig. 2b,e. Energy-dispersive X-ray spectroscopy (EDX) sulfur mapping in Supplementary Fig. 2h, corresponding to the SEM images in Supplementary Fig. 2g, with top view revealed homogeneous distribution of DBU-ILs over the thimble of SiNWs. In particular, the SEM-tilted view and the corresponding sulfur mapping in Fig. 2a,b confirmed selective immobilization of DBU-ILs, [HDBU$^+$][MIm$^-$] catalyst, only on the tips of SiNWs. In addition, the presence of fluorine element from the [HDBU$^+$][TFE$^-$] catalyst was also proven by EDX spectra (Supplementary Fig. 13). After immobilization of ILs selectively in the form of thimbles, the CA of DMSO and water solvents became very low ($\sim 10°$) due to hydrophilic nature of the immobilized ILs. Because of the low CA, the interface between gas and liquid would move slightly down into the SiNWs cluster below the thimbles of immobilized ILs, as illustrated in Fig. 2c (ref. 27), presumably due to Cassie–Baxter state of wetting behaviour. This lowering of the interface was observed (Fig. 2d) when the ILs-immobilized sample was dipped into a mixture of DMSO solution containing a fluorescent dye (4,4-difluoro-1,3,5,7-tetra-methyl-4-bora-3a,4a-diaza-s-indacen: BODIPY 505/515). To confirm a stable gas–liquid laminar flow with direct interface contact, dyed DMSO (10 µl min$^{-1}$) was injected into the upper part of the channel, and nitrogen (75 µl min$^{-1}$) into the channel lower part filled with SiNWs along 80 cm long microchannel. Bubbles were

not observed in the liquid flow and no dyed DMSO was seen in the gas outlet (Supplementary Movie 1, Supplementary Fig. 14).

**Synthesis through $CO_2$ capture and fixation.** The catalytic performance and efficacy of DBU-ILs immobilized SiNWs microreactor were evaluated by conducting two kinds of gas–liquid interface reactions: $CO_2$ addition with propargylic amines and that with 2-aminobenzonitriles to synthesize 2-oxazolidinones and quinazoline-2,4-(1H,3H)-diones, respectively, which are an important class of compounds for the synthesis of medicines in pharmaceutical industries and organic synthesis. The reactions catalysed by [HDBU$^+$][MIm$^-$] were studied first between pure or diluted $CO_2$ and propargylic amines for the synthesis of 2-oxazolidinones, by infusing a DMSO solution of N-butyl-1-phenylhex-1-yn-3-amine (1a) (0.04 M) as a model compound of propargylic amines (Supplementary Figs 15–28) to the upper part and $CO_2$ gas to the lower SiNWs part of the channel of the DBU-IL immobilized microreactor using leak-proof syringe. In general, the reaction performance depended on the flow rate (residence time), the concentrations of reactant solution and $CO_2$, as summarized in Table 1.

Under molar ratios of 5.6–8.4 ($CO_2$/amine), 98% yield of 5-benzylidene-3-butyl-4-propyloxazolidin-2-one (2a) was obtained in 188 s of residence time at 40 °C and ambient pressure in the presence of 0.02 mmol ILs catalyst (Table 1, Entry 3), resulting in $\sim 2.5$ mmol h$^{-1}$ productivity. In addition, the additional immobilization of DBU-IL catalyst on the modified PDMS channel contributed to a slight increase in the yields from 94 to 98% (Table 1, Entry 4). It clearly indicates that the major reaction must have occurred by the interfacial catalyst on the thimbles of SiNWs, while only minor contribution was made by the catalyst on the modified PDMS wall, which acted on the diffused $CO_2$ gas through the solution.

After optimization of 2-oxazolidinones synthesis with pure $CO_2$ gas, a natural gas containing 2.95 mol% $CO_2$ was chosen as a model of flue gas to test the reaction under different molar ratios

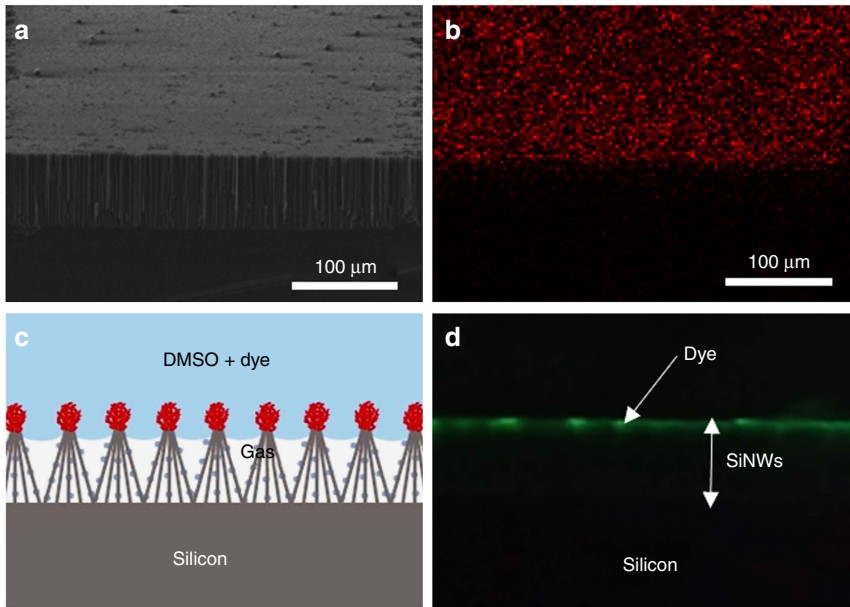

**Figure 2 | DBU-ILs immobilization on the SiNWs.** (**a**) Tilted SEM image of selective mercaptosilane immobilized SiNWs, Scale bar, 100 μm and (**b**) corresponding sulfur EDX-mapping. Scale bar, 100 μm. (**c**) Schematic illustration of gas-green dye solution interface (Cassie–Baxter state) over the ILs immobilized SiNWs and (**d**) corresponding fluorescence image after dipping into the BODIPY dye-dissolved DMSO solution, visualizing the actual interfacial position of liquid/gas in the ILs immobilized SiNWs clusters. Scale bar, 100 μm.

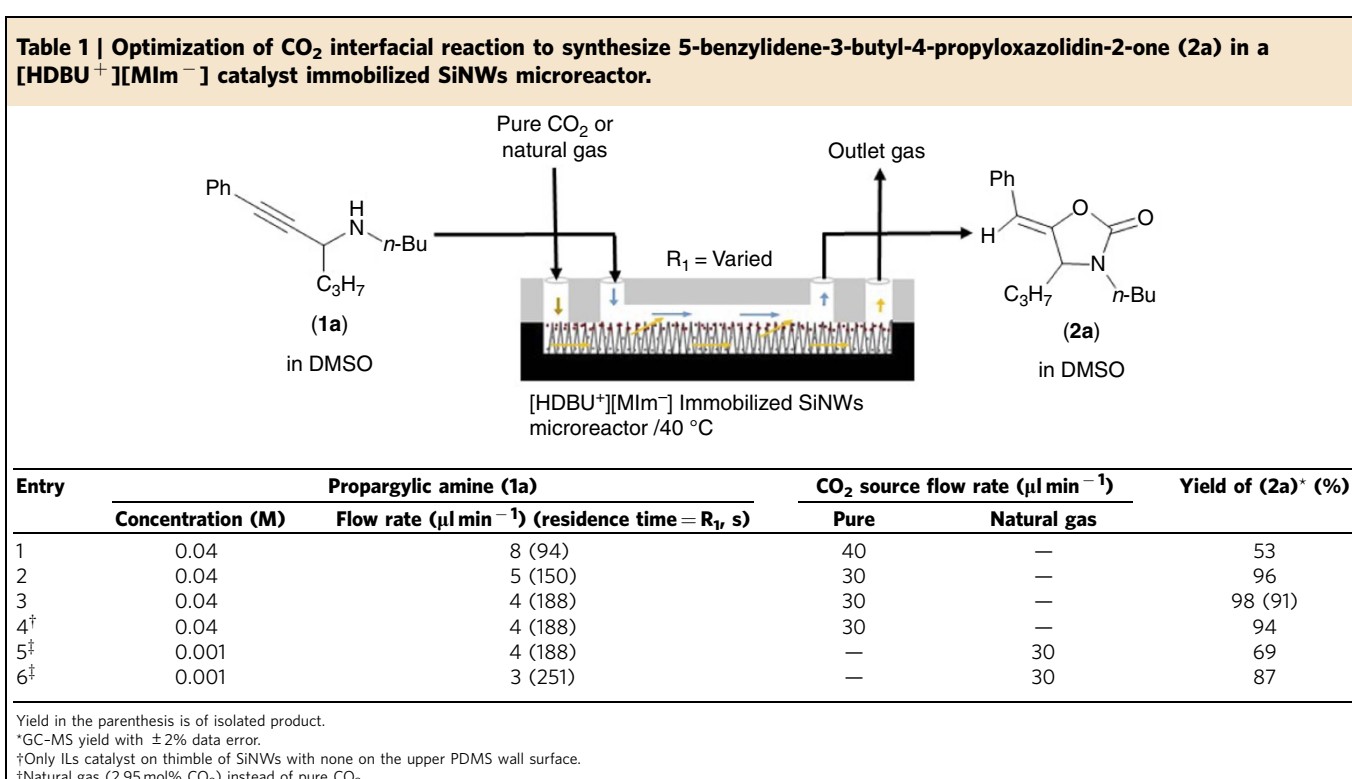

**Table 1 | Optimization of $CO_2$ interfacial reaction to synthesize 5-benzylidene-3-butyl-4-propyloxazolidin-2-one (2a) in a $[HDBU^+][MIm^-]$ catalyst immobilized SiNWs microreactor.**

| Entry | Propargylic amine (1a) | | $CO_2$ source flow rate ($\mu l\,min^{-1}$) | | Yield of (2a)* (%) |
|---|---|---|---|---|---|
| | Concentration (M) | Flow rate ($\mu l\,min^{-1}$) (residence time = $R_1$, s) | Pure | Natural gas | |
| 1 | 0.04 | 8 (94) | 40 | — | 53 |
| 2 | 0.04 | 5 (150) | 30 | — | 96 |
| 3 | 0.04 | 4 (188) | 30 | — | 98 (91) |
| 4[†] | 0.04 | 4 (188) | 30 | — | 94 |
| 5[‡] | 0.001 | 4 (188) | — | 30 | 69 |
| 6[‡] | 0.001 | 3 (251) | — | 30 | 87 |

Yield in the parenthesis is of isolated product.
*GC–MS yield with ±2% data error.
†Only ILs catalyst on thimble of SiNWs with none on the upper PDMS wall surface.
‡Natural gas (2.95 mol% $CO_2$) instead of pure $CO_2$.

of 9.8–13.16 ($CO_2$/amine). It took a longer reaction time (251 s) to attain a reasonable yield (87%) due to slower reactions in lower concentrations of reagent and $CO_2$. When the collected natural gas from the outlet of microreactor was analysed by GC–MS, the $CO_2$ content in the natural gas was decreased by ∼0.34% (2.61% from 2.95%) by adsorption in liquid medium (0.15 mol%) and consumption for chemical conversion (0.19 mol%), which is consistent with the calculated amount from mass balance, *ca.* 1.0 ($CO_2$/amine), with no interfering effect of other gases in the reaction (Supplementary Fig. 29a–d). Presumably it is feasible to capture $CO_2$ from other flue gas containing higher $CO_2$ contents by controlling the reaction conditions including flow rate parameter. In addition, nearly no catalytic degradation or leaching occurred even after 4 days of continuous reaction,

**Table 2 | Optimization of $CO_2$ interfacial reaction to synthesize quinazoline-2,4-(1$H$,3$H$)-dione (3a) in a [HDBU$^+$][TFE$^-$] catalyst immobilized SiNWs microreactor.**

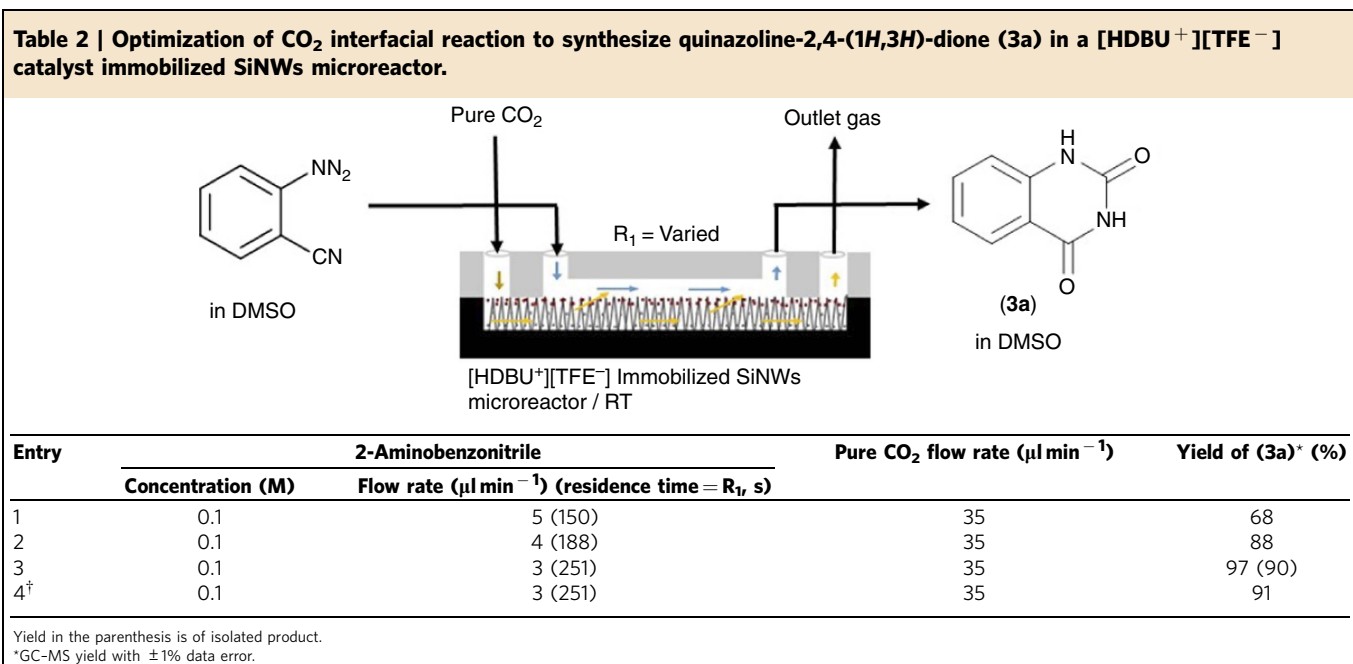

| Entry | 2-Aminobenzonitrile | | Pure $CO_2$ flow rate ($\mu$l min$^{-1}$) | Yield of (3a)* (%) |
|---|---|---|---|---|
| | Concentration (M) | Flow rate ($\mu$l min$^{-1}$) (residence time = R$_1$, s) | | |
| 1 | 0.1 | 5 (150) | 35 | 68 |
| 2 | 0.1 | 4 (188) | 35 | 88 |
| 3 | 0.1 | 3 (251) | 35 | 97 (90) |
| 4$^\dagger$ | 0.1 | 3 (251) | 35 | 91 |

Yield in the parenthesis is of isolated product.
*GC–MS yield with ±1% data error.
$^\dagger$Only ILs catalyst on thimble of SiNWs with none on the upper PDMS wall surface.

delivering the yield in the range of 98–90%, which is an indication of high stability of immobilized DBU-IL catalyst (Supplementary Fig. 30).

The identical methodology was applied for [HDBU$^+$][TFE$^-$] catalysed reactions of 2-aminobenzonitrile derivatives to synthesis quinazoline-2,4-(1$H$,3$H$)-diones as summarized in Table 2. The reaction performance generally depended on the residence time of 2-aminobenzonitrile at fixed flow rate of $CO_2$ gas. In the range of molar ratio of 3.1–5.2 ($CO_2$/2-aminobenzonitrile), the optimized reaction resulted in 97% conversion (Table 2, Entry 3) within 4.2 min of residence time at room temperature, resulting in ∼2.8 mmol h$^{-1}$ productivity. In contrast, the conventional batch process required a longer reaction time (24 h) and a larger [HDBU$^+$][TFE$^-$] catalyst loading (∼6 times higher than the substrate concentration) to attain ∼95% conversion from pure $CO_2$ source[20]. There is also nearly no catalytic degradation or leaching even after 4 days (Supplementary Fig. 31). The excellent reaction and capturing performance in the presence of DBU-ILs immobilized catalyst can be attributed to highly efficient mass transfer via direct contact between the liquid reagent flowing in the upper part of the channel and the underlying $CO_2$ gas stream through the SiNWs. Furthermore, the microfluidic approach taken here exploits the intrinsic advantages of high surface to volume ratio, and fast heat and mass transfer capability, which leads to accelerated reaction kinetics in the presence of minimal catalyst loading as reported in various organic synthesis[28–32].

**Autonomous serial process of reaction and work-up.** The concept depicted earlier in Fig. 1 for a fully autonomous serial process of reaction and work-up can now be applied to 2-oxazolidones synthesis, as shown in Fig. 3. Due to high boiling point (189 °C) of DMSO solvent, it was troublesome to work-up by evaporating the solvent for product isolation under vacuum. To make easy work-up procedure we have switched low volatile DMSO solvent to highly volatile dichloromethane (DCM, boiling point 40 °C) after the reaction[21]. In this work, therefore, liquid–liquid extraction of droplet microfluidics is adopted to isolate the synthesized 2-oxazolidones in DCM from the DMSO mixture,

followed by separation of the DCM mixture using a phase microseparator embedded with hydrophobic PTFE membrane (Supplementary Figs 32 and 33)[30,33,34]. It is well known that water is well miscible with DMSO to become a hydrophilic aqueous-like phase, but immiscible with hydrophobic DCM medium. To induce a sufficient difference in the polarity of DMSO/water and DCM for conducting droplet-based extraction, water was first infused at different flow rates to the DMSO mixture coming from the SiNWs microreactor at T1 junction (Fig. 3) to form a homogeneous DMSO/water phase in the PTFE capillary for 2.2 min, and formed droplets by merging with DCM at T2 to accomplish the extraction of both reagent and product from the DMSO/water phase for 2.7 min (details in Supplementary Methods and Supplementary Table 2). This autonomous separation was achieved in the microseparator by a simple principle that the hydrophobic DCM containing product could preferentially wet and penetrate through the PTFE membrane, while the non-wetting DMSO/water is unable to penetrate the PTFE membrane (Supplementary Movies 2 and 3).

Figure 3 reveals that the fully automated serial process with ∼0.02 mmol of reusable catalyst can deliver various 2-oxazolidinone products in excellent yields over 90% at 40 °C for substituted propargalic amines (2a–2f) in 8 min of total process time: 3.1 min for $CO_2$ capturing/fixation reaction, 2.2 min, and 2.7 min for water mixing and extraction in DCM, respectively (detailed procedure in Supplementary Methods). In contrast, the conventional flask process required a higher temperature (60 °C), and a longer reaction time (8–26 h) even with a larger loading of [HDBU$^+$][MIm$^-$] catalyst (≥2 equiv. of propargylic amine)[19], which needs to be separated and reloaded for reuse of the catalyst.

The calculated turnover number from the microreactor was found to be more than 12 times higher than that of the conventional batch process. The identical methodology was further applied, this time with $CO_2$ in natural gas in place of pure $CO_2$, to give an excellent yield of 83% for 2-oxazolidinone products (Fig. 3, 2a*). The $^1$H and $^{13}$C NMR spectra of all purified 2-oxazolidinones products are shown in Supplementary Figs 35–46.

The autonomous total process of $CO_2$ conversion reaction and work-up was extended to the synthesis of various quinazoline-

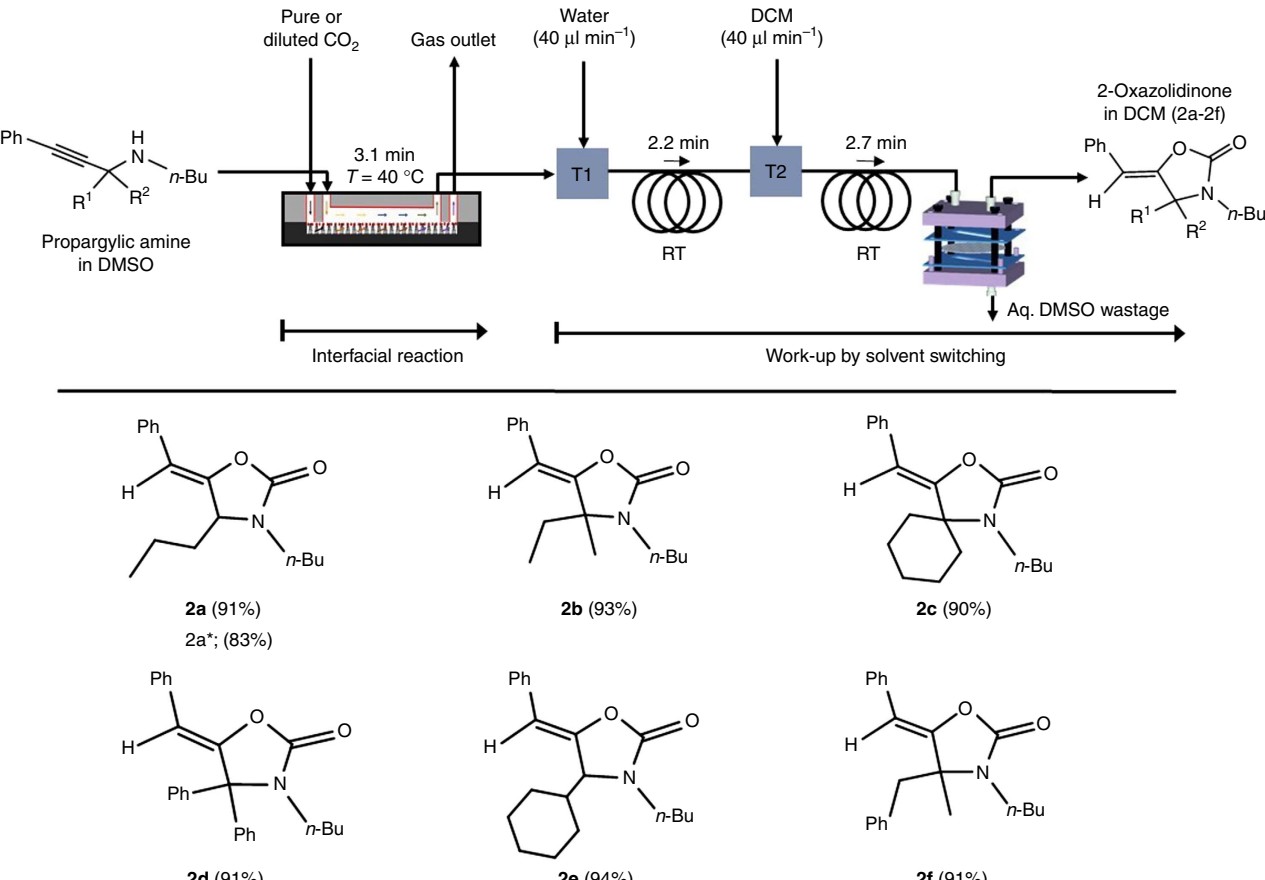

**Figure 3 | Synthesis of 2-oxazolidinones.** 2-Oxazolidinones synthesis in the autonomous integrated serial process of $CO_2$ interfacial [HDBU$^+$][MIm$^-$] catalytic reaction (188 s with pure $CO_2$, 251 s with natural gas containing 2.95 mol% $CO_2$) as optimized in Table 2, and subsequent solvent switching (DMSO to DCM) for facile work-up. Yields in the parenthesis are of isolated products and data error limit ± 2.0% is acceptable.

2,4-(1H,3H)-dione derivatives in the [HDBU$^+$][TFE$^-$]-ILs immobilized microreactor, followed by a work-up step for separation of the products (Fig. 4). The introduction of DCM flow (flow rate 20 μl min$^{-1}$) into the DMSO reaction medium containing the substrate and product led to the precipitation of the products, (quinazoline-2,4-(1H,3H)-diones), due to the insolubility, while the soluble unreacted substrates (2-aminobenzonitrile derivatives), remained in DCM (detailed procedure in Supplementary Methods). All the reactions proceeded very smoothly to give excellent yields over 81% by simple filtration (Fig. 4, 3a–3g). The $^1$H and $^{13}$C NMR spectra of all purified quinazoline-2,4-(1H,3H)-diones products are shown in Supplementary Figs 47–60.

## Discussion

In summary, we have presented an integrated platform for fully autonomous *in situ* $CO_2$ capturing/fixing and simple work-up to synthesize and isolate high value-added chemicals. The platform with ionic liquid catalysts immobilized on the tips of bundled SiNWs has been shown to produce and isolate the desired products in excellent yields under mild conditions. The super-amphiphobic nature of the SiNWs allows a stable continuous interface between a liquid reagent flow and a gas phase containing $CO_2$ that flows in parallel with the upper liquid flow, resulting in efficient catalysed $CO_2$ gas–liquid reactions in the continuous laminar flow. This on-site $CO_2$ capture and *in situ* conversion concept has successfully been extended to a natural gas

containing 2.95 mol% $CO_2$ as a model of flue gas, demonstrating the possibility of applying the platform for direct air capture or natural gas purification.

## Methods

**General.** Silicon wafer (100) (p-type, dopant: boron) was purchased from Semi-Materials Co., Ltd. AZ 1512 positive photoresist was purchased from AZ Electronic Materials. SU-8 negative photoresist was purchased from Micro Chem. Polydimethylsioxane (PDMS, Sylgard 184) and curing agent (Sylgard 184) were purchased from Dow Corning. Allylhydridopolycarbosilane (AHPCS, SMP-10) was purchased from Starfire System. PTFE (id = 500 μm) tubing and T-junction were purchased from Upchurch Scientific. Co. HF solution (≥48 wt%) was purchased from Sigma-Aldrich. $H_2O_2$ solution (30 wt%) was purchased from Samchun Pure Chemical Co. Ltd. Calorimetric natural gas reference standard was purchased from Sigma-Aldrich. All organic solvents and reactants from Sigma-Aldrich or Alfa Aesar or TCI chemicals were used as received without any additional purification. For experiments the deionized water with conductivity 18.2 mS was used. The chemical structures of synthesized products were fully characterized by GC/Mass-spectra, $^1$H and $^{13}$C NMR. GC/MS spectrum was recorded by Agilent 5975C GC/MSD System. $^1$H NMR, $^{13}$C NMR and HSQC NMR spectrum were recorded on a Bruker 600 or 300 MHz in either CDCl$_3$ or DMSO-$d_6$. High-resolution SEM images were taken by Philip XL30 SEM, operating at $10^{-2}$ to $10^{-3}$ Pa with EHT 15 kV with 300 V collector bias. Platinum sputtering was implemented at pressure ranging between 1 and 0.1 Pa prior to SEM experiments. CAs were measured using Smart Drop (FemtoFab). The etched fraction of SiNWs was analysed by calculating the ratio of etched area to the whole area of SiNWs from the SEM image through Image J software.

**Typical procedure for synthesis of 2-oxazolidinone (2a).** A stock solution of N-butyl-(1-phenylethynyl-butyl)-amine (1a) (0.04 M) was prepared by dissolving 0.916 g of 1a in 100 ml DMSO. The reaction was proceeded by infusing a solution of 1a (flow rate 4–8 μl min$^{-1}$) to the upper PDMS channel and pure $CO_2$ (flow

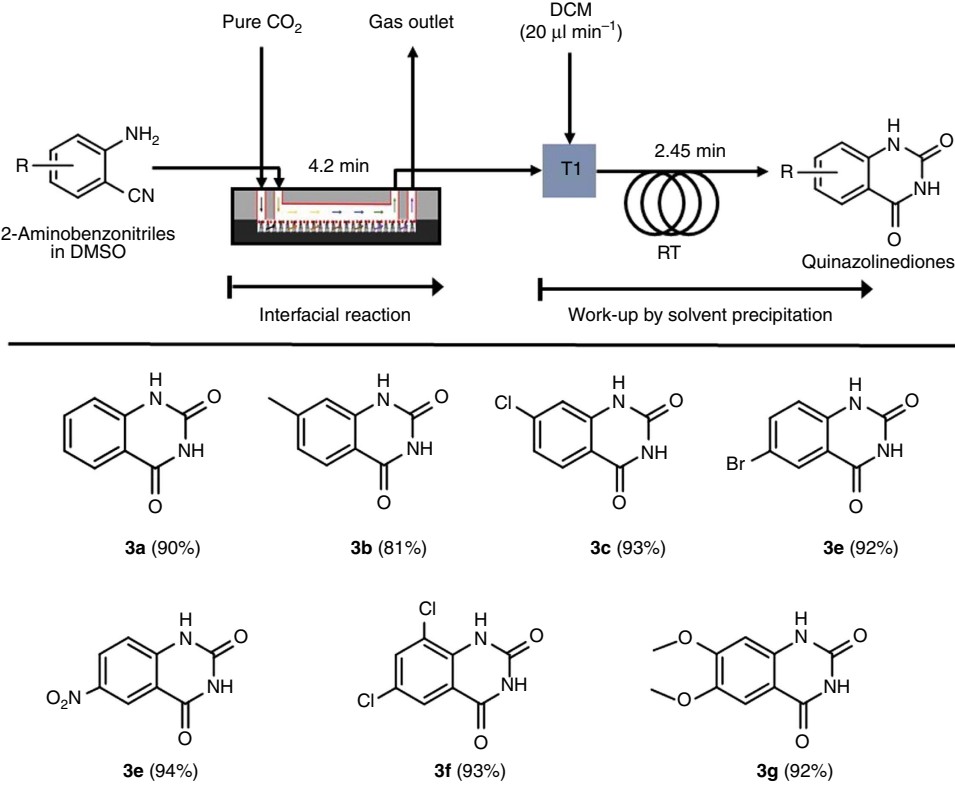

**Figure 4 | Synthesis of quinazoline-2,4-(1H,3H)-diones.** Quinazoline-2,4-(1H,3H)-diones synthesis in the autonomous integrated serial process of $CO_2$ interfacial [HDBU$^+$][TFE$^-$] catalytic reaction for 251 s with pure $CO_2$ as optimized in Table 2, and subsequent precipitation by adding DCM. Yields in the parenthesis are of isolated products and data error limit ± 2.0% is acceptable.

rate 30–50 µl min$^{-1}$) to the lower SiNWs channel of microreactor using individual leak proof syringe (Supplementary Fig. 28). Residence time of reactant and flow rate of gas were varied to obtain the product with different yields determined by GC–MS. Best conversion (98%) was achieved at 4 µl min$^{-1}$ flow rate (188 s retention time) of 1a (Table 1, Entry 3). Molar ratio of $CO_2$ to propargylic amine, 1a, was varied from 5.6 to 8.4 while in the case of natural gas molar ratio varied from 9.8 to 13.16. Remaining gas was collected from the outlet of microreactor, and analysed by GC–MS. The $CO_2$ contents in the natural gas was decreased ~0.34% (2.61 from 2.95%) by chemical adsorption by DMSO solvent ($\gamma_1$) and reactant/product ($\gamma_2$) (0.15 mol%) and consumption for chemical conversion (0.19 mol%), which is consistent with the calculated amount from mass balance, ca. 1.0 ($CO_2$/amine) (Supplementary Fig. 29). Details calculation is given in Supplementary Methods.

**Typical synthesis of quinazoline-2,4(1H,3H)-dione (3a).** A stock solution of 2-aminobenzonitrile (0.1 M) in DMSO was prepared. The reaction was proceeded by infusing a solution of 2-aminobenzonitrile (flow rate 3–5 µl min$^{-1}$) to the upper PDMS channel and pure $CO_2$ gas with flow rate 30–50 µl min$^{-1}$ to the lower SiNWs channel of microreactor using individual leak proof syringe. Residence time of reactant and flow rate of gas were varied to obtain the product with different yields determined by GC–MS. The highest yield (97%) was achieved at 3 µl min$^{-1}$ flow rate at 251 s retention time of 2-aminobenzonitrile (Table 2, Entry 3). After completion of reaction, DCM was infused to the outlet of the microreactor. The product (quinazoline-2,4(1H,3H)-dione) was precipitated in DCM to obtain by filtering, while reactant (2-aminobenzonitrile) was dissolved to be drained out. Molar ratio of $CO_2$ to 2-aminobenzonitrile was varied from 3.1 to 5.2 as calculations are given in Supplementary Methods.

**Data availability.** The data that support the findings of this study are available in the article or its Supplementary File, or available from the corresponding author on request.

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

## Acknowledgements

We gratefully acknowledge the support from the National Research Foundation (NRF) of Korea grant funded by the Korean government (NRF-2008-0061983).

## Author contributions

N.K.V. and A.K.S. contributed equally by conducting overall experiments and the design, data analysis. D.-H.K. designed the SiNW microreactor and helped SiO$_2$ coating over the SiNWs. Y.-H.H. conducted contact angle measurement. J.-O.K. helped wax coating. A.G.B. conducted fluorescence measurement experiment. N.K.V., A.K.S. and D.-P.K. wrote the manuscript. D.-P.K. conceived the project.

## Additional information

**Competing financial interests:** The authors declare no competing financial interests.

**Publisher's note**: 

