## [Peer review file · Nature Communications]

Reviewers' comments:

Reviewer #1 (Remarks to the Author):

This paper describes an interesting specially designed microfluidic system for the reaction of organic compounds with CO₂. The reactor design enables a stable gas/liquid laminar flow, and the catalyst is appropriately located at the interface of gas and liquid. The reactor design shown here is noble and unique, and was found to be quite efficient for fixing CO₂ into organic molecules. This particular method should provide very exciting breakthroughs in this field.

I have some minor points.

1. It might be nice if the authors provide information about the effect of the laminar width of gas and liquid. If the interface does not fit the position of the catalyst, what happens?

2. It is also nice if the authors indicate the productivity of the system. How many moles or mmoles of the product can be produced per unit time?

Overall, I strongly support publication of the manuscript.

Reviewer #2 (Remarks to the Author):

This work reported a novel interfacial catalytic reaction platform for an integrated autonomous process of simultaneously capturing/fixing CO₂ in gas- liquid laminar flow and subsequently providing a work-up step in series. The platform with reported DBU-ILs catalysts immobilized on the tips of bundled SiNWs was shown to display the significantly improved efficacy for CO₂ conversion under mild conditions. The paper gave interesting results, and was organised well. It deserves to publish in this journal.

Reviewer #3 (Remarks to the Author):

The manuscript submitted by Kim et al. details an investigation into the development of a microfluidic reactor that promotes interfacial catalysis between a liquid and gas phase. A key feature of the platform is laminar gas-liquid flow, achieved through the fabrication of functionalised silicon nanowires within the reactor microchannel. An innovative aspect of this approach is the immobilization of an ionic liquid (IL) catalyst at the tip of the bundled silicon nanowires, providing a stable gas-liquid interface for efficient gas (CO₂) transport into the reaction zone. A benefit of this approach is that the immobilized IL catalyst does not mandate post-reaction separation and work-up, which is a limitation of ILs catalysts within this context.

The microfluidic reactor design builds on previous work by the same group who pioneered protocols for the immobilization of silicon nanowires within a microchannel (Nature Commun., DOI: 10.1038/ncomms10741). The authors have subsequently applied the reactor technology to address the challenge the utilization of CO₂ renewable C1 resource for fine chemical manufacture. The effectiveness of this platform was assessed through the

study of two transformations. The synthesis of 2-oxazolidinones and quinazolinones from the reaction of CO₂ with propargylic amines, and 2-aminobenzonitriles, respectively. The synthesis of derivatives from each of these class of heterocycle was well tolerated in the reactor platform, affording most isolated products in high yield. Another feature of the works relates to the use of in-line production (solvent switching and extraction) to generate products of high purity without additional work-up procedures.

Overall the manuscript is reasonably well written and data presented in a logical, systematic fashion. However, the manuscript has a number of deficiencies:

1. Introductory remarks – there are numerous sweeping and unsubstantiated statements claimed by the authors, which include; (i) Page 2, Line 30. How is ‘huge emission’ of CO₂ defined? Statistics supporting an estimate of global CO₂ emissions as a result of human activity should be included, as this forms the basis of justification of the research in the manuscript. (ii) Page 2, Line 33. It is unclear how the authors have assessed and defined ‘market demands’? (iii) Page 2, Line 49. Again it is unclear how the authors have assessed and defined ‘process efficiency’

2. Novelty of the approach. (i) The fabrication of the reactor platform was recently reported in the same journal (*vide supra*), with the exception of the surface immobilization of the IL catalysts, this is almost identical to previously published work. The immobilization of ILs to supports is also well established (e.g. H.-J. Choi et al. / *Chemical Engineering Science* 100 (2013) 242–248). (ii) The IL catalyzed synthesis of 2-oxazolidinones and quinazolinones propargylic amines, and 2-aminobenzonitriles with CO₂ has been previously reported by Han et al. (*Angew. Chem. Int. Ed.* 2015, 54, 5399–5403) and Liu et al. (*Angew. Chem. Int. Ed.* 2014, 53, 5922 –5925). Each of these previous papers reports the use of a 1,8-diazabicyclo[5.4.0]undec-7-ene (DBU) IL to promote CO₂ sequestration and the corresponding reaction. The authors should consider a broader scope of heterocycle synthesis (or alternative organic molecules) to demonstrate utility and novelty of the approach, particularly given the potential application to the fine chemical industry. I would also advise the authors to consider accessing novel reactivity with the reactor platform to permit the synthesis of new heterocycles.

3. Scalability and Industrial Translation. The authors suggest that the reactor platform could be deployed to an industrial scale to convert CO₂ waste streams as feedstocks for fine chemicals manufacture. The reactor platform is clearly a prototype, however no suggestion is made on how this technology could be translated into the industrial setting. An economically viable process would require the processing of millions of tonnes of CO₂ from a range of flue gas sources. The fabrication of the reactor hierarchical catalyst within larger scale flow reactors could be prohibitively expensive. Furthermore, the process utilizes environmentally incompatible and expensive solvents (DMSO and dichloromethane), which may also prohibit the uptake of the technology by industry. These factors combine to unfortunately reduce the potential and broader impact of the science.

In consideration of the above points, it is my opinion that the work is promising, however the manuscript does not currently meet the requirements for publication in *Nature Communications*. The above should be considered to help improve the manuscript to engage the broader interest of the readership and novelty of the work.

Reviewer #1 (Remarks to the Author):

This paper describes an interesting specially designed microfluidic system for the reaction of organic compounds with CO₂. The reactor design enables a stable gas/liquid laminar flow, and the catalyst is appropriately located at the interface of gas and liquid. The reactor design shown here is noble and unique, and was found to be quite efficient for fixing CO₂ into organic molecules. This particular method should provide very exciting breakthroughs in this field.

I have some minor points.

1. It might be nice if the authors provide information about the effect of the laminar width of gas and liquid. If the interface does not fit the position of the catalyst, what happens?

Reply: The microreactor is composed of the bottom channel with built-in SiNWs (83 cm length × 500 μm width × 70 μm height) and the upper channel with surface modified PDMS (80 cm length × 500 μm width × 30 μm height). Therefore, the laminar width/depth of gas and liquid must be the same as the dimensions of the microchannel, and the different dimension was not attempted. However, the direct interface contact below the thimbles of immobilized DBU-ILs in the gas-liquid laminar flow was proven by fluorescent dye test (Figure 2f) as well as no cross-flow between gas and liquid over 80 cm long channel, when varied the infusion rate of the solutions (3-10 μL min⁻¹) and the CO₂ gas (30-75 μL min⁻¹) into the microreactor. Eventually, the specific positioning of catalysts at the binary interface only enabled excellent yields (81-97%) under mild conditions.

2. It is also nice if the authors indicate the productivity of the system. How many moles or mmoles of the product can be produced per unit time?

Reply: The productivity of 5-benzylidene-3-butyl-4-propyloxazolidin-2-one (2a) is ~2.5 mmol h⁻¹, which was originally mentioned in the text (page 9, line 3) based on Entry 3 of Table 1(a) result. While in the case of quinazoline-2,4-(1H,3H)-diones (3a) productivity, is found ~2.8 mmol h⁻¹ within 4.2 min of residence time based on (Table 1(b), Entry 3) result, has been newly added in text (page 10, line 4).

Overall, I strongly support publication of the manuscript.

Reviewer #2 (Remarks to the Author):

This work reported a novel interfacial catalytic reaction platform for an integrated autonomous process of simultaneously capturing/fixing CO₂ in gas-liquid laminar flow and subsequently providing a work-up step in series. The platform with reported DBU-ILs catalysts immobilized on

the tips of bundled SiNWs was shown to display the significantly improved efficacy for CO₂ conversion under mild conditions. The paper gave interesting results, and was organised well. It deserves to publish in this journal.

Reviewer #3 (Remarks to the Author):

The manuscript submitted by Kim et al. details an investigation into the development of a microfluidic reactor that promotes interfacial catalysis between a liquid and gas phase. A key feature of the platform is laminar gas-liquid flow, achieved through the fabrication of functionalised silicon nanowires within the reactor microchannel. An innovative aspect of this approach is the immobilization of an ionic liquid (IL) catalyst at the tip of the bundled silicon nanowires, providing a stable gas-liquid interface for efficient gas (CO₂) transport into the reaction zone. A benefit of this approach is that the immobilized IL catalyst does not mandate post-reaction separation and work-up, which is a limitation of ILs catalysts within this context.

The microfluidic reactor design builds on previous work by the same group who pioneered protocols for the immobilization of silicon nanowires within a microchannel (Nature Commun., DOI: 10.1038/ncomms10741). The authors have subsequently applied the reactor technology to address the challenge the utilization of CO₂ renewable C1 resource for fine chemical manufacture. The effectiveness of this platform was assessed through the study of two transformations. The synthesis of 2-oxazolidinones and quinazolinones from the reaction of CO₂ with propargylic amines, and 2-aminobenzonitriles, respectively. The synthesis of derivatives from each of these class of heterocycle was well tolerated in the reactor platform, affording most isolated products in high yield. Another feature of the works relates to the use of in-line production (solvent switching and extraction) to generate products of high purity without additional work-up procedures. Overall the manuscript is reasonably well written and data presented in a logical, systematic fashion. However, the manuscript has a number of deficiencies:

1. Introductory remarks – there are numerous sweeping and unsubstantiated statements claimed by the authors, which include;

(i) Page 2, Line 30. How is ‘huge emission’ of CO₂ defined? Statistics supporting an estimate of global CO₂ emissions as a result of human activity should be included, as this forms the basis of justification of the research in the manuscript.

Reply: “huge emission” has been replaced by “tens of gigatons emission per year” with the corresponding references (Ref. 1, Nature 526, 628-630, (2015), Ref. 2 PLoS ONE 8, e81648, (2013)) as newly added.

(ii) Page 2, Line 33. It is unclear how the authors have assessed and defined ‘market demands’?

Reply: This part has been rephrased as, “However, these technologies have some inherent drawbacks including low CO₂ capacity, solvent loss, equipment corrosion, high cost for regeneration (Ref. 6, Nat. Commun. 6, 6124, (2015)).”

(iii) Page 2, Line 49. Again it is unclear how the authors have assessed and defined ‘process efficiency’

Reply: The part has been elaborately defined by revising as following: “the highly viscous ionic liquid suffered from low gas-to-liquid mass transfer in addition to insufficient surface-to-volume ratio and heat transfer of conventional batch reactor. Therefore, the limited process efficiency required.....”

2. Novelty of the approach.

(i) The fabrication of the reactor platform was recently report in the same journal (vide supra), with the exception of the surface immobilization of the IL catalysts, this is almost identical to previously published work. The immobilization of ILs to supports is also well established (e.g. H.-J. Choi et al. / Chemical Engineering Science 100 (2013) 242–248).

Reply: There are lots of reports on SiNWs for electronic and optical applications at literatures. Our previous works have extended the use into the gas-liquid laminar flow of microreactors for homogenous catalytic reactions and evaporative separation with different principles, respectively. On the other hand, this work presents an interfacial catalytic reaction platform for simultaneously capturing and fixing CO₂ in the SiNW-embedded mciroreactor where the key concept is to selectively position IL catalysts on the SiNWs tips at the binary interface of gas-liquid laminar flow, enabling excellent yields for several minutes of reaction under mild conditions and repeated use of the catalyst. Furthermore, a subsequent work-up step is provided to isolate the desired synthesized products in the continuous-flow manner. However, the work (H.-J. Choi et al./Chem. Eng. Sci., 100, 242-248, (2013)) reported the ILs immobilization onto polyethylene glycol (PEG) support for catalytic activity in a batch reactor at conventional conditions (120 °C, 1.3-1.9 MPa for 4 h, precipitating recovery of the catalyst), failing to overcome the limitations of the ordinary CO₂-based process.

(ii) The IL catalyzed synthesis of 2-oxazolidinones and quinazolidinones propargylic amines, and 2-aminobenzonitriles with CO₂ has been previously reported by Han et al. (Angew. Chem. Int. Ed. 2015, 54, 5399-5403) and Liu et al. (Angew. Chem. Int. Ed. 2014, 53, 5922 –5925)> Each of these previous papers reports the use a 1,8-diazabicyclo[5.4.0]undec-7-ene (DBU) IL to promote CO₂ sequestration and the corresponding reaction. The authors should consider a broader scope of heterocycle synthesis (or alternative organic molecules) to demonstrate utility and novelty of the approach, particularly given the potential application to the fine chemical industry. I would also advise the authors to consider accessing novel reactivity with the reactor platform to permit the synthesis of new heterocycles.

Reply: This work mainly claimed the new invention of interfacial catalytic reaction platform for an integrated autonomous process of simultaneously capturing/fixing CO₂ in gas-liquid laminar flow with subsequent work-up step for facile isolation of products. The usefulness of the developed platform was demonstrated by producing and isolating 2-oxazolidinones and quinazolidinones with high yields under mild conditions and no catalyst recovery. In this respect, this work showed the innovative process against the reports by Han et al. and Liu et al. (cited as Ref. 19, 20) that took long reaction time and high catalyst loading (6 h & 2 times for 2-oxazolidinones, 24 h & 6 times for quinazolidinones, respectively) and followed the routine steps of work-up and catalyst recovery. Therefore, it is plausible that this platform would enable to produce high-valued specialty chemicals such as novel heterocycles directly from flue gases in the future.

3. Scalability and Industrial Translation. The authors suggest that the reactor platform could be deployed to an industrial scale to convert CO₂ waste streams as feedstocks for fine chemicals manufacture. The reactor platform is clearly a prototype, however no suggestion is made on how this technology could be translated into the industrial setting. An economically viable process would require the processing of millions of tonnes of CO₂ from a range of flue gas sources. The fabrication of the reactor hierarchical catalyst within larger scale flow reactors could be prohibitively expensive. Furthermore, the process utilizes environmentally incompatible and expensive solvents (DMSO and dichloromethane), which may also prohibit the uptake of the technology by industry. These factors combine to unfortunately reduce the potential and broader impact of the science.

Reply: All authors modestly accept that this continuous-flow reaction platform showed only small scale in academic purpose by use of costly organic solvents (DMSO and DCM) with high surface tension. However, it is assured that this innovative membrane-free approach would encourage new

attempts to overcome these limitations by developing the superomniphobic structured surface with facile technology as well as the use of green and economic solvents like water to perform CO₂-based C1 chemistry, which would be great for employing in industry level.

In consideration of the above points, it is my opinion that the work is promising, however the manuscript does not currently meet the requirements for publication in Nature Communications. The above should be considered to help improve the manuscript to engage the broader interest of the readership and novelty of the work.

REVIEWERS' COMMENTS:

Reviewer #1 (Remarks to the Author):

The manuscript has been revised satisfactorily and I recommend publishing it.

Reviewer #3 (Remarks to the Author):

The authors have prepared an adequately revised manuscript which now highlights the novelty of the work; the fabrication of a microreactor with selective positioning of an ionic liquid catalyst at the tip of silicon nanowire tips enabling stable gas/liquid laminar flow within the reactor microchannel. In consideration of the revisions, I recommend the paper be considered for publication in Nature Communications.